# Multisystem recovery after sport-related concussion in adolescent rugby players: a prospective study protocol

Connor Shane McKee ,[1] Mark Matthews,[2] Alan Rankin ,[3,4] Chris Bleakley[1]

¹School of Health Sciences, Ulster University, Belfast, UK
²School of Sport, Ulster University, Belfast, UK
³Sports Medicine NI, Belfast, UK
⁴Sports Medicine, Sports Institute of Northern Ireland, Newtownabbey, UK

**Correspondence to**
Prof Chris Bleakley;
c.bleakley@ulster.ac.uk

## ABSTRACT

**Introduction** Sport-related concussion is one of the most common injuries in adolescent rugby players with evidence of prolonged recovery in some concussed athletes. Concussion is a complex pathophysiological process that can affect a variety of subsystems with multifactorial presentation. Most research on adolescents recovery after concussion focuses on neurocognitive functioning and symptom outcomes over the short term. There is a need to explore concussion recovery over time across multiple subsystems in adolescent rugby players.

**Methods and analysis** This prospective study will use sensorimotor and oculomotor outcomes in adolescent male and female rugby players aged 16–18 years. Players will be recruited from school or club rugby teams across the province of Ulster. Baseline assessment will be undertaken at the start of the playing season and will include questionnaires, Quantified Y Balance Test and Pupillary Light Reflex. Players who sustain a concussive event will be reassessed on all outcomes at 3 days, 7 days, 14 days, 23 days, 90 days, 180 days and 365 days postconcussion. For serial outcome data, we will examine response curves for each participant and make comparisons between known groups. We will use logistic regression to explore any association between demographic variables and recovery. The strength of the predictive model will be determined using $R^2$, p values and ORs, with 95% CIs.

**Ethics and dissemination** Ethical approval has been granted for this study from Ulster University Research Ethics Committee (REC/14/0060). This study will be published in an open-access research journal on completion.

**Trial registration number** ACTRN12622000931774p.

## STRENGTHS AND LIMITATIONS OF THIS STUDY

⇒ Female adolescent rugby players have been overlooked in adolescent concussion research, this study seeks to address this with a large sample of both male and female adolescent rugby players.
⇒ Many studies cease follow-up within 6 weeks postconcussion, this study adopts a prospective design which aims to track recovery of sport-related concussion for 12 months.
⇒ This study uses innovative, portable and relatively low-cost technology, which may be feasible for use in a non-professional, youth sporting environment.
⇒ Despite the growing popularity of female rugby in Northern Ireland, there is a limited population of female adolescent rugby players, in particular aged 16–18 years, to recruit from.
⇒ Access to players at key follow-up dates may be challenging given the limited access to adolescent rugby players during school hours at an important examination stage for this population.

## INTRODUCTION

Sport-related concussion (SRC) is a complex pathophysiological process affecting the brain induced by traumatic biomechanical forces.[1] Paediatric visits to emergency departments for concussion and minor head injuries have doubled in the USA in recent years.[2] Many concussions occur during contact and collision sports like rugby union. The rate of concussion in professional rugby union is estimated at 21.5 per 1000 game hours.[3] Incidence is also high in adolescent rugby with estimates of an incidence of 6.1 concussions per 1000 game hours, which is equivalent to 3 concussions per team per season.[4]

SRC can affect a variety of subsystems, including oculomotor,[5] somatosensory,[6] vestibular systems,[7] with disturbances to cognition,[1] mood[8] and biomarker profile.[9] Clinical recovery after an SRC can typically take between 10 days and 14 days in adults[1] and up to 4 weeks in adolescents.[10] It is recommended adolescent athletes undergo a battery of tests postconcussion,[1 11–13] including the Sports Concussion Assessment Tool.[14] While research into the recovery from concussion in adolescents is growing, most research focuses primarily on neurocognitive functioning, subjective and symptom-based outcomes.[15–17] Few prospective studies examine sensorimotor or oculomotor function postconcussion, and the natural progression of recovery for these subsystems is poorly understood.

The sensorimotor system plays a central role in regulating postural control and balance. Retrospective data suggest that 30%

of adolescents presenting with concussion have vestibulo-ocular deficits in the acute postinjury period.[18] Adolescent players with acute vestibular dysfunction also have worse prognosis[19] with a fourfold greater risk of developing postconcussion syndrome.[20] Clinical assessment of sensorimotor function is based on balance tests such as balance error scoring system (BESS), dynamic tasks like the Y Balance Test, or a 3-metre walk test. However, these traditional tests are scored subjectively, and lack the responsiveness needed to detect small but important deficits in sensorimotor performance. Adding wearable sensor technology to traditional balance tests can improve their accuracy, allowing researchers to quantify subclinical balance deficits. Previous research has shown that sensor-based instrumentation of the Y Balance Test provides a reliable and valid measure of movement control,[21 22] and is more responsive than the traditional reach distance scoring method.[23] In a study of adolescent rugby union athletes aged 14–18 years, those with a history of concussion had poorer dynamic balance (high sample entropy (SEn)) during Y Balance Tests compared with non-concussed controls.[24] There is evidence of Quantified Y Balance Tests predictive validity, and elite male rugby union players presenting with greater irregularity in their Y Balance Test movement control were three times more likely to sustain a concussion than the players with more regular Y Balance Test movement control.[23]

Oculomotor dysfunction commonly occurs following SRC.[19] This can manifest through symptoms of blurred vision and sensitivity to light. Abnormalities have been observed in the pupillary light reflex (PLR) following concussion.[25] PLR is underpinned by a complex autonomic function, where pupillary constriction is dominated by the parasympathetic pathway, and dilation dominated by the sympathetic system. Traditionally, PLR was examined subjectively using a penlight test, where the practitioner shines a light into a patient's eyes; but this lacks diagnostic accuracy and results can be confounded by changes in attention, accommodation and environmental ambient light. Pupillometry is a diagnostic technology offering a standardised and objective evaluation of the PLR, providing a quick, accurate and quantifiable evaluation of the PLR[25] and capturing a range of outcome parameters across different phases of PLR response (eg, peak and average dilatation/constriction velocity). Normative PLR values are documented in adults, but these are affected by age,[26] and initial data suggest that adolescents (aged 12–18 years), and particularly males, show greater maximal pupil diameters, slower maximum constriction velocities and lower percent constriction.[26] These developmental differences make translating the results of adult studies to children difficult. Recently, PLR technology has been able to accurately display meaningful differences between adolescent athletes with a concussion and those without on eight out of nine PLR metrics,[25] but no longitudinal data have been reported.

Evidence suggests that maladaptive coping strategies such as fear-avoidance behaviours relating to an injury have also emerged as potentially important prognostic factors in predicting prolonged recovery.[27] Fear-avoidance behaviours have increased in relevancy to concussion recovery in adolescents.[28 29] It would be plausible fear-avoidance and anxiety behaviours around potential or perceived injuries could be associated with prolonged symptoms. Recent studies have highlighted fear-avoidance behaviours were positively correlated with somatic symptoms, emotional distress and negatively associated with quality of life in a study involving adolescent subjects with concussion,[30] which supports findings from adult samples.[28 29] There is a need to explore fear-avoidance behaviours, anxiety and depressive symptoms further in the recovery from SRC in adolescents.

SRCs in adolescents remain a great concern due to the uncertainty surrounding their pathophysiology, the potential moderating effects of cognitive maturation and poorly understood recovery. New technologies (eg, wearable inertial sensors and pupillometry) can enhance the existing battery of assessment, providing more objective, natural recovery data for key subsystems (oculomotor, sensorimotor). Sex differences exist throughout development, but there is a dearth of prospective research involving adolescent females who sustain an SRC and recovery. There is a need for longitudinal research in SRC as many neurophysiological deficits persist beyond the point of clinical symptom resolution.[31–35]

## Aims

The aim of this study is to prospectively study recovery from an SRC in adolescent male and female rugby players. The objectives of the study are to (1) examine if sensorimotor and oculomotor outcomes are affected by concussion history, sex and age and (2) reassess sensorimotor and oculomotor outcomes and self-reported measures at time points postconcussion over a 12-month period.

## Study design

This study comprises a cross-sectional and prospective component, with follow-up continuing over a single playing season. Ethical approval has been granted from Ulster University Research Ethics Committee (REC/14/0060).

## Patient and public involvement

Participants and/or the public were not involved in the design, or conduct, or reporting, or dissemination plans of our research.

## Participants

Participants will be recruited from school or club rugby teams across the province of Ulster. There are more than 30 school teams with male adolescent players, and 25 registered clubs with dedicated female youth teams across Ulster. Many of these sites have previously been active members of the Rugby Injury Surveillance in Ulster Schools and have successfully collaborated in previous studies.[4 24 36] To be eligible for inclusion, participants must be (1) aged between 16 years and 18 years, (2) injury

free at the time of recruitment and (3) currently playing rugby union in the 2022/23 playing season.

Consenting participants will complete a baseline questionnaire covering demographic characteristics (date of birth, sex, weight, height, concussion history). Parental consent is not required as the participants are aged between 16 years and 18 years and can provide their own assent for participation. The researcher team will be present throughout to assist with the understanding and completion of the questionnaire.

## METHODS AND ANALYSIS
### Outcome measures
The outcome measures will be a range of physical assessments and self-reported questionnaires, including symptoms and psychological measures as these are often involved in SRC. All outcome measures will be taken at 3 days, 7 days, 14 days and 21 days and 3 months, 6 months and 12 months postconcussive event.

### Physical assessment
Quantified Y Balance Test is a modification of the Star Excursion Balance Test.[37] A detailed testing protocol has previously published.[24] Participants position themselves on a platform with hands firmly on their hips. They are instructed to use their non-stance leg to slide a box forward as far as possible with their foot and return to the starting upright position in three defined directions (anterior, posteromedial and posterolateral). Reach distances are recorded to the nearest 0.5 cm. Individuals complete three practice trials, followed by three recorded trials in a randomised order. A failed attempt is noted when any of the following errors occurred: touching of the foot down on the floor before returning to the starting position; placing the foot on top of the sliding box for balance; and flicking or kicking the sliding box for a better performance or any loss of balance.[38] If an individual meets any of the failure criteria, the reach attempt will be discarded and repeated. During all tests, an inertial sensor (Shimmer3; Shimmer Sensing) is mounted at the level of the fourth lumbar vertebra and secured with a custom-made elastic belt to closely match the acceleration of the body's centre of mass during the Y Balance Test excursions. Inertial sensor testing will be captured during testing and analysed offline with MATLAB (2018b; MathWorks). Reach distances and the Y Balance Test will be normalised in relation to the individuals' leg length, and the mean of three trials completed on each direction will be obtained. Movement control during the Y Balance Test will be quantified using the following variables: normalised reach distance and the SEn of the three axes of the gyroscope signal (x, y and z). SEn of the signal of length N=(x1, x2, x3, …, xn), will be calculated using the following formula:

Sample Entropy = −log (A/B)

Our interpretation of SEn scores is underpinned by key theories in the optimal variability during human motor performance, described by Stergiou and Decker, where a low SEn score indicates more regular and less complex movement control during the Y Balance Test reach excursions, while a higher SEn score indicates a higher degree of movement irregularity/complexity.[39]

PLR metrics will be measured in response to a brief, step-input, white light stimulus (154-millisecond duration; 180-microwatt power) via a Neuroptics PLR-3000 handheld, infrared, automated, monocular pupillometer (Neuroptics). This device is United States Food and Drug Adminstration (FDA) approved and has been used in similar studies of mTBI in adults.[40] The pupillometer captures dynamic responses 32 times per second, analysing a continuous, 5-second digital video of the pupillary response to light. The metrics quantified by the device software include the following: maximum pupil diameter (steady-state pupil size before the light stimulus), minimum pupil diameter (pupil size after maximum constriction in response to the light stimulus), percentage pupil constriction, latency (time to maximum constriction in response to the light stimulus), peak and average constriction velocity, peak and average dilation velocity and T75 (time for pupil redilation from minimum diameter to 75% maximum diameter). Peak dilation velocity will be calculated from automated slope-based measures obtained by the pupillometer (Microsoft Excel 2016; Microsoft).

Trained research staff will conduct PLR assessments in an environment with a room illumination of approximately 350 lx (moderate photopic viewing conditions). Athletes will focus on a 3-metre distance target with the non-tested eye for ocular fixation and accommodation during a 5-second measurement period. Monocular measurements are to be repeated at least three times for each eye, alternating 1-minute time intervals to allow rapid visual light adaptation, to obtain two to three artifact-free responses per eye. The combined mean of each pupillometry metric is then calculated for at least two assessments without artefacts, defined as blinks or eye movements occurring within the first 3 s of the response.

### Self-report questionnaires
The Post-Concussion Symptom Scale (PCSS) consists of 22 questions that relate to postconcussive symptoms. Users are asked to rate each symptom according to a 7-point Likert Scale ranging from 0 to 6. Higher scores indicate a higher severity of postconcussive symptoms. The greatest possible score is 132 and the lowest possible score is 0. The scale has a high internal consistency[41] in healthy and concussed adolescents, its construct validity has been established[42] and it can accurately discriminate between concussed and non-concussed athletes.[41]

The Paediatric Fear Avoidance Behaviour after Traumatic Brain Injury Questionnaire (PFAB-TBI) is a 16-item measure designed to assess fear avoidance following paediatric concussion developed based on FAB-TBI. The FAB-TBI was developed to assess fear-avoidance behaviour postconcussion in adults. Items are related to avoidance

of activities (eg, work) due to fear of symptom consequences or symptom exacerbation (eg, worsened headache). For the PFAB-TBI self-report item content will remain the same with the exception of two items. Item 4 will be changed from 'My work might harm my brain' to 'My schoolwork might hurt my brain', and item 5 will be changed from 'I should not do my normal work with my present symptoms' to 'I should not do my normal schoolwork with my present symptoms'. The PFAB-TBI asks respondents to rate agreement on statements related to avoidance behaviour over the past month from 0 (strongly disagree) to 3 (strongly agree). A total sum score will be used in the current study, where higher scores indicate greater fear-avoidance behaviour (possible range 0–48). Change scores will be calculated across testing dates.[30]

The Generalised Anxiety Disorder (GAD-7) is a 7-item questionnaire that has questions about possible problems the respondent may have been bothered by over the previous 2 weeks. Each question has four possible responses from 'not at all' to 'nearly every day' and scored 0–3, respectively. The GAD-7 has reported good reliability, as well as criterion, construct, factorial and procedural validity.[43]

The Patient Health Questionnaire (PHQ-9) is a self-reported questionnaire with nine questions about experienced depression symptoms. Questions cover interest in things, feelings of hopelessness, sleep, fatigue, appetite, feeling like a failure, restlessness, concentration and thoughts about suicide. Each question is scored from a '0' (not at all) to '3' (nearly every day) and has been validated for primary care use.[44]

The Sport-Related Concussion Clinical Profiling Questionnaire consisting of 29 questions covering possible symptoms and severity of symptoms a player may have experienced over the past 24 hours. Each question has a 4-point scale ranging from 'no symptoms experienced' to 'severe' and scored 0–3 for each question. The questionnaire explores five possible clinical profiles including cognitive/fatigue, vestibular, ocular, migraine and anxiety/mood and two modifiers, sleep and neck pain.[45]

### SRC events

SRC prevalence will be recorded over a single playing season (2022/23). SRC definition will follow the criteria set out by the Zurich Consensus Statement on Concussion in Sport[1]; that is, a direct blow to the head, face, neck or elsewhere on the body with an impulsive force transmitted to the head resulting in one or more of the symptoms in one or more of the following clinical domains (which may or may not have involved loss of consciousness): somatic, cognitive, emotional/behavioural and sleep disturbance. All SRCs must be verified by a medical professional within 48 hours of the injury. When a participant is diagnosed with a concussion, the designated person at each school/club (data champion) will be responsible for contacting the primary researcher (CSM). CSM will maintain weekly contact with the data champion, sending study reminders via text message or a follow-up phone call. The following

information will be inputted for each concussion: date of injury, reporter/person making the diagnosis, site where diagnosis was made, cause of injury (phase of play, direct vs indirect), time to first symptoms and nature of first symptoms.

In the event of an SRC, all outcome measures will be reassessed by the primary researcher after 3 days, 7 days, 14 days and 21 days and 3 months, 6 months and 12 months. Dates of return to training and return to play will be self-reported by the participant or coach. Return to play dates will be cross referenced with the official team lists registered with Ulster Rugby. The researchers involved in this study will have no influence on the return to training or return to play decisions; these decisions will remain external to the current project. Participants will be asked on return visits as to whether they have experienced any treatment or intervention relating to their concussion between visits. Any and all interventions undertaken by individuals will be considered and reported when analysing the results.

### Statistical analysis

All data will be analysed using the Statistical Package of Social Sciences (SPSS) (V.26; SPSS).

Baseline outcomes (concussion history, Y Balance, PLRM) will be summarised using means and SDs for scale variables and frequencies and percentages for nominal variables. We will use histograms to examine the frequency distributions of scale variables, and where normality is assumed, we will use independent t-tests to compare baseline data (PCSS, Y Balance, PLRM), with players categorised into two groups based on self-reported history of concussion in the last 12 months (no concussion in the last 12 months or concussion within the last 12 months). To avoid bias, researchers will be blinded to group title (ie, history of concussion vs no history of concussion). The threshold for statistical significance will $p < 0.01$ for all tests. We will also present mean differences (95% CIs) between groups for scale outcome data.

SRCs occurring throughout the 2022/23 playing season will be summarised using counts (n, %). Time to SRC event will be plotted using survival curves, with participants censored at the end of the playing season. Serial outcome data (PCSS, Y Balance, PLRM, PFAB-TBI) will be measured at 72 hours, 7 days, 14 days, 23 days, 90 days, 180 days and 365 days postconcussion; we will initially determine the number (%) of participants that had returned to baseline at each time point. We will also examine the growth/response curves for each participant, as well as compare curves between known groups; sex and concussion history. As the shape of the time-response curve is uncertain and may vary across each outcome variable, we will calculate summary measures for each participant based on either the peak value, the time to return to baseline or the area under the curve. Area under the curve will be calculated using the trapezium rule (by adding the areas under the graph between each pair of consecutive observations). The most suitable summary measure will

be used in the calculation of a median score (IQRs) for each outcome variable. Between-group comparisons will also be made, comparing curves across two known groups, for example, males/females and history of concussion/no history.

In an exploratory analysis, we will use logistic regression to examine the association between demographic variables (age, sex, previous injury) and recovery at 23-day follow-up. We will run separate models for each outcome measure (PCSS, Y Balance, PLRM, PFAB-TBI). Our previous research[4] found that 90% of adolescent rugby players return to sport at 23 days postconcussion; this is also the minimal convalesce for adolescent players postconcussion based on Irish Rugby Football Union and World Rugby guidance, which are built around the latest medical research and expert-based opinion.[46] Full recovery will be defined as a return to baseline (preseason) values for each respective outcome. Initially, we will conduct a series of univariate analyses to determine whether any predictor variables are associated with recovery. Correlations among predictor variables will be calculated to screen for any strong colinearity (r>0.8). Predictors demonstrating a p value less than 0.10 on univariate testing will be entered into a multiple logistic regression analysis. The strength of the predictive model will be determined using $R^2$, p values and ORs, with 95% CIs.

## Sample size

We aim to enrol 250 participants at baseline. This figure allows for a 20% loss to follow-up over the entire study and assumes a 10% concussion prevalence over a single playing season. This will generate recovery data from approximately 20 participants presenting with a new concussion. Recruitment will be stratified to include equal numbers of male and female participants (n=125 males; n=125 females).

A sample of 250 participants would also provide sufficient power for detecting baseline difference in sensorimotor function. Our previous research[24] had a medium effect standardised mean difference (SMD) 0.5 between adolescents with a history of concussion to those with no history for sensorimotor function with SDs of 0.22 units (SEn). Based on an alpha level of 1%, n=180 participants would provide 90% power for detecting a medium effect difference in SEn between groups at baseline.

## DISCUSSION

The primary aim of this study is to study recovery from an SRC in adolescent male and female rugby players and whether this is affected by concussion history, sex and age. The ability to confidently diagnose a concussion and monitor its resolution in adolescents has proven somewhat challenging. For an initial SRC diagnosis in adolescents, no single, gold-standard test exists,[47] and even widely accepted tests have questionable reliability, such as the modified BESS, with which groups have been found to have high variability and large number of errors.[48] Multiple areas of SRC sign and symptom endorsement should be evaluated, which requires an individualised approach to each athlete and injury.[47] Given the absence of a diagnostic test or biomarker for concussion in adolescents, the current cornerstone of concussion diagnosis is confirming the presence of a constellation of signs and symptoms with multiple tests after an individual has experienced a hit to the head or body although existing evidence is insufficient to determine the best combination of measures.[49]

A mix of quantitative and qualitative metrics has been chosen to identify and track recovery from concussion including several self-report measures. Despite the documented importance of reporting concussive symptoms, research has suggested that up to half of concussions in adolescent football players are unreported and therefore undiagnosed.[50] The mixed-methods approach employed in the proposed research may allow for an interesting view of whether adolescent rugby players under-report symptoms when quantitatively measured metrics are still elevated.

Women's rugby is growing in popularity, but there is little evidence available regarding injuries and in particular recovery from concussion.[51] Research has highlighted that they experience worse outcomes associated with concussion[52] and are more likely to experience a concussion when compared with males[53] and this highlights an area of focus for our research.

This trial protocol aims to optimise methodological quality and report pragmatic clinical findings by addressing key methodological limitations of other studies that have aimed to track recovery of concussion in adolescents. Key strengths of the study include the following: (1) a large even sample population of adolescent male and female rugby players, (2) participant stratification into predetermined subgroups based on predetermined criteria (ie, concussion history), (3) multiple outcome measures to assess a wide array of deficits associated with SRC and (4) longitudinal follow-up to explore possible persistence of multisystem deficits associated with SRC.

## ETHICS AND DISSEMINATION
### Ethics approval and consent to participate

Participating in this study poses little to no risk to participants or their parents. All patients and their parents/guardians will provide written informed consent/assent and will have the ability to withdraw at any time without explanation. Participation in the study will not restrict them from receiving the standard diagnosis and management, as determined by treating physicians. We will ensure that data remain secure, and that patient privacy and confidentiality are maintained.

Ethical approval has been granted through the Ulster University Research Ethics Committee (REC/14/0060).

## Dissemination

The findings of this research will be published in a peer-reviewed open access medical journal. All positive, inconclusive or negative findings will be disseminated to maximise the social value of the research and to accurately inform future practises. This will occur as soon as possible after completion of the final analysis.

**Contributors** All authors contributed to the concept, design and writing of this protocol. CSM, MM, AR and CB each had substantial contributions to the conception, design, planning, as well as the acquisition, analysis and interpretation of data for the work. CSM drafted the initial content, and MM, AR and CB reviewed it critically for important intellectual content, suggesting and making changes to the work to ensure the highest quality. CSM, MM, AR and CB collectively approved the final version to be published. CSMK, MM, AR and CB were in agreement to be accountable for all aspects of work in ensuring that questions related to the accuracy or integrity of any part of the work were appropriately investigated and resolved.

**Funding** This research is to be completed during a PhD funded by the Department for the Economy at Ulster University. These provide for the payments of approved fees and the maintenance of students while being trained in methods of PhD research as such there is no specific award/grant number for this project from my funder that I can provide. The funding body will have no impact on the study design, data collection, analysis, interpretation of data or in writing the manuscript.

**Competing interests** None declared.

**Patient and public involvement** Patients and/or the public were not involved in the design, or conduct, or reporting, or dissemination plans of this research.

**Patient consent for publication** Not applicable.

**Provenance and peer review** Not commissioned; externally peer reviewed.

**Open access** This is an open access article distributed in accordance with the Creative Commons Attribution 4.0 Unported (CC BY 4.0) license, which permits others to copy, redistribute, remix, transform and build upon this work for any purpose, provided the original work is properly cited, a link to the licence is given, and indication of whether changes were made. See: https://creativecommons.org/licenses/by/4.0/.

**ORCID iDs**
Connor Shane McKee http://orcid.org/0000-0002-0831-8481
Alan Rankin http://orcid.org/0000-0001-5132-1937

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
