## [Reviewer comments · BMJ Open]

ARTICLE DETAILS

TITLE (PROVISIONAL)	Multi-system recovery after sports-related concussion in adolescent rugby players: A prospective study protocol.
AUTHORS	McKee, Connor; Matthews, Mark; Bleakley, Chris; Rankin, Alan

VERSION 1 – REVIEW

REVIEWER	Roberts, William O University of Minnesota Twin Cities
REVIEW RETURNED	14-May-2023

GENERAL COMMENTS	The manuscript clearly outlines your plan. I have included some editing suggestions in the attached PDF of your manuscript. - The reviewer provided a marked copy with additional comments. Please contact the publisher for full details.
---

REVIEWER	Conder, Robert Carolina Neuropsychological Service Inc
REVIEW RETURNED	15-May-2023

GENERAL COMMENTS	The study proposal is comprehensive and thorough. Reflects an up to date understanding of sport concussion, especially issues with teenage females. I especially appreciate the inclusion of biometric measures, as they may reflect both ANS & CNS correlates of sport concussions. Here's the issue: how do you account for treatment and intervention, rather than reflecting the naturalistic course of sport concussions over time ? I don't know the status of sport concussion treatment by healthcare professionals in the UK, but in the US, these athletes would receive therapeutic intervention. This would alter the data findings, especially the post-acute findings. This needs to be addressed.
--

VERSION 1 – AUTHOR RESPONSE

Comment 1: Please clearly state that this is a study protocol in the Title.

Response: We have updated the title so that it includes study protocol. The new title is: "Multi-system recovery after sports-related concussion in adolescent rugby players: a prospective study protocol."- Page 1, line 2-3.

Comment 2: Please reformat the abstract so that it follows the structured abstract recommended in the journal's instructions for authors for study protocols.

Response: We have reformatted the abstract so that it meets the journal's outlined abstract guidelines- Page 1, line 10-32.

Comment 3: We note that you state that the "study protocol follows the STROBE guidelines". Please note that the STROBE guidelines relate to the reporting of research articles, not protocols. As such, you may want to consider revising this statement.

Response: We have removed this statement.

Comment 4: Please include the subheading "patient and public involvement" above your patient and public involvement statement in the main text.

Response: We have added the subheading "patient and public involvement" above our patient and public involvement statement in the main text. Page 4, line 142.

Comment 5: Please reformat the main text so that it follows the structure recommended in the journal's instructions for authors for study protocols, for example the main text of your manuscript should contain an Ethics and Dissemination section.

Response: We have re-formatted the main text so that it follows the structure recommended in the journal's instructions for authors for study protocols. An ethics and dissemination section has been added to the main text. See page 10, line 373-388

Comment 6: Here's the issue: how do you account for treatment and intervention, rather than reflecting the naturalistic course of sport concussions over time ? I don't know the status of sport concussion treatment by healthcare professionals in the UK, but in the US, these athletes would receive therapeutic intervention. This would alter the data findings, especially the post-acute findings. This needs to be addressed.

Response: Active rehabilitation at an early stage is the gold standard of concussion care. In the UK, active treatment is not readily accessible outside elite sport. It would usually not be accessed until symptoms become prolonged, beyond the time-frames of this study and unfortunately unlikely to be a confounding issue. Participants will be asked on return visits as to whether they have experienced any treatment or intervention relating to their concussion between visits. Any and all interventions undertaken by individuals will be considered and reported when analysing the results. This statement has now been included on page 7, line 274-277.

Comment 7: This does not make sense to me. Do you mean ... with evidence of prolonged recovery in some concussed athletes.

Response: Yes this is what we meant. I have updated it to read 'with evidence of prolonged recovery in some athletes' on page 1, line 12.

Comment 8: Focuses

Response: Changed the word focus to focuses- change has been made on page 1, line 14.

Comment 9: Do you mean r-squared? Missing superscript

Response: Yes I meant r-squared. Change has been made on page 1, line 28.

Comment 10: no need for caps and consider and odds ratios (OR) with 95%....

Response: I have removed the capitals on page 1, line 28-29.

Comment 11: than adult players

Response: added in 'than adult players', but then removed as it was in the discussion section of the abstract which is not required for BMJ Open, hereby this was removed.

Comment 12: strike through text 'across a playing season, and for up to 12 months post-concussion'

Response: removed this

Comment 13: Do not include words that are in the title so you have space for other key words

Response: Removed original keywords and added new ones on page 2, line 51-52. New keywords are as follows: Female, Ocular, Cognitive, Vestibular, Sensorimotor, Anxiety, Fear, Balance, Pupillary Light-Reflex

Comment 14: delete comma

Response: deleted comma after brain on page 2, line 56

Comment 15: Consider: Many concussions occur during contact and collision sports like rugby union

Response: accepted the change, see line page 2, 59

Comment 16: deleted 'and prospective research' and 'an', suggested replacing with 'with' and 'of an'

Response: accepted the change, see page 2, line 61

Comment 17: deleted 'changes to'

Response: accepted the change and deleted 'changes to', see page 2, lines 65-66

Comment 18: deleted 'but can be protracted for'

Response: accepted the change and deleted 'but can be protracted for' and replaced this with 'and', see page 2, line 67

Comment 19: deleted 'should'

Response: accepted the change and deleted 'should', see page 2, line 68.

Comment 19: deleted 'but not exclusively limited to'

Response: accepted the change and deleted 'but not exclusively limited to', see page 2, line 69.

Comment 20: changed word 'remains' to 'is'

Response: Accepted the change and removed the word 'remains' and replaced it with 'is', see page 3, line 73.

Comment 21: Removed a comma

Response: Removed the comma, see page 3, line 77.

Comment 22: non-concussed controls?

Response: Accepted the change and change 'healthy control' to non-concussed controls. See page 3, line 88.

Comment 23: strike through 'who possessed' and changed to 'with'

Response: Accepted the change and inserted word 'with'. See page 3, line 91.

Comment 24: removed a comma

Response: removed this in the updated text. See page 3, line 101.

Comment 25: Strike through '. It provides' and changed to 'providing'

Response: changed to providing. See page 3, line 102.

Comment 26: Strike through 'captures' and changed to 'capturing'

Response: accepted change, changed to 'capturing'. See page 3, line 103.

Comment 27: Suggested changing word 'fewer' to 'lower'

Response: Accepted the change. See page 3, line 107.

Comment 28: suggested adding word 'and'

Response: Accepted the change, and added the word 'and' on page 4, line 136.

Comment 29: strike through 'registered'

Response: Removed registered, see page 4, line 148.

Comment 30: is parental consent required?

Response: Parental consent is not required as the participants are aged between 16-18 and can provide assent for their own participation. Updated in text, see page 5, line 155-157.

Comment 31: not sure if the comma should precede or follow whereby

Response: changed it to in front of whereby. See page 5, line 189.

Comment 32: I think these can all be commas

Response: changed from semi-colons to commas. See page 6, line 198-203

Comment 33: removed a comma

Response: Removed comma in the text. See page 6, line 204.

Comment 34: changed 'self-reported questionnaires' to 'self-report questionnaires'

Response: Accepted the change, see page 6, line 214.

Comment 35: capitalize Likert

Response: Accepted the change, capitalized Likert. See page 6, line 216.

Comment 36: deleted 'covers' and changed to 'has'

Response: Accepted change, see page 7, line 236.

Comment 37: deleted 'covering' and changed to 'with'

Response: Accepted change, see page 7, line 242.

Comment 38: deleted 'depressive' and changed to 'depression'

Response: Accepted change, see page 7, line 243

Comment 39: Insert text 'about suicide'

Response: Added the text 'about suicide', see page 7, line 244

Comment 40: removed a comma

Response: Removed comma, see page 7, line 249

Comment 41: removed word 'respectively'

Response: removed word 'respectively', see page 7, line 251

Comment 42: added a comma

Response: Added a comma, see page 7, line 253

Comment 43: added word 'pain' after 'neck'

Response: added word 'pain' after 'neck', see page 7, line 253

Comment 44: add a semi colon after 'Zurich Consensus Statement on Concussion in Sport'

Response: added a semi colon after 'Zurich Consensus Statement on Concussion in Sport', see page 7, line 257

Comment 45: removed a comma after the word 'head'

Response: Removed a comma after the word head, see page 7, line 258

Comment 46: removed a semi-colon and replaced with ', and' after word behavioural

Response: removed a semi-colon and replaced with ', and' after word behavioural, see page 7, line 260

Comment 47: Remove the sentence 'the primary researcher'

Response: removed the sentence, 'the primary researcher' and replaced with 'CMcK' see page 7, line 264

Comment 48: Removed brackets around 'CMcK'

Response: Removed brackets around 'CMcK', see page 7, line 264

Comment 49: Add in 'n' alongside '%'

Response: Added in 'n' alongside '%', see page 8, line 292

Comment 50: I'm not a fan of etc. Can you include the other comparisons?

Response: This was a mistake, the two comparison groups are simply concussion history and sex, there are no other groups. I have removed 'etc.' and replaced between 'sex' and 'concussion history' with 'and'. See page 8, line 298

Comment 51: Add a semi colon after 'known groups'

Response: Added a semi colon after 'known groups', see page 8, line 298

Comment 52: Add a comma after 'e.g.'

Response: Added a comma after 'e.g.', see page 8, line 305

Comment 53: Is this data based or expert based recommendation?

Response: added a reference to the IRFU return to play protocol which indicates that it follows the World Rugby Guidelines for return to play decisions. World Rugby's guidelines are 'built around the latest medical research and expert based opinion.' I have added this into the main text- see page 8, line 312-313

Comment 54: Remove comma after 'post-concussion'

Response: Removed comma after 'post-concussion', see page 8, line 312

Comment 55: change formatting of R2 to R²

Response: Changed R2 to R². See page 8, line 319

Comment 56: Removed capital letters from Odds Ratios

Response: Removed capital letters from odds ratios. See page 8, line 319

Comment 57: Remove n= and replaced with approximately

Response: Removed n= and replaced with approximately, see page 9, line 324

Comment 58: Removed comma

Response: Removed comma after 0.5, see page 9, line 329

Comment 59: Removed comma

Response: Removed comma after 'history', see page 9, line 330

Comment 60: Strike through ', via' and replaced with 'with'

Response: Replaced 'via' with 'with', see page 9, line 344

Comment 61: Most of this has already been said and may be redundant here

Response: Removed this paragraph, see page 9, line 353-354

Many thanks for the comments and feedback,

Connor McKee